# Graph Personalized Federated Learning via Client Network Learning

**Jiachen Zhou**                                                                 *jz3668@columbia.edu*
*Department of Computer Science*
*Columbia University*

**Xie Han**                                                                       *han.xie@emory.edu*
*Department of Computer Science*
*Emory University*

**Carl Yang**                                                                    *j.carlyang@emory.edu*
*Department of Computer Science*
*Emory University*

**Reviewed on OpenReview:** *https://openreview.net/forum?id=pyTTR4pxkU*

## Abstract

Graph classification is a widely studied problem for applications such as molecule/protein function prediction and drug discovery. Powerful graph neural networks (GNNs) have demonstrated state-of-the-art performance for the classification of complex graphs, but training such models can require significant amounts of high-quality labeled graphs that are expensive to collect. When individual institutes do not possess sufficient graph data, federated learning (FL) becomes a handy solution for them to collaboratively obtain powerful graph models without directly sharing their own graph data. However, existing FL frameworks for graph data do not consider the realistic setting of personalized FL with heterogeneous data, where each client aims to leverage the data of certain other clients to boost its own model performance. In this work, inspired by graph structure learning, we propose to learn a dynamic client network that tracks the graph data similarity across clients to guide model sharing along FL. Specifically, we rely on the marginal parameters of local GNNs to dynamically learn the client network, and refer to a set of fundamental graph properties to guide its learning. Extensive experiments on three real-world graph datasets demonstrate the consistent effectiveness of our two major proposed modules, which also mutually verify the effectiveness of each other.

## 1 Introduction

Federated Learning (FL) has gained widespread popularity as a machine learning paradigm that enables distributed model training without sharing local data samples McMahan et al. (2017); Li et al. (2020). A significant challenge that limits the performance of FL is data heterogeneity across clients. As diverse local data collected by different clients in various scenarios often adhere to non-identical distributions Zhao et al. (2018); Zhu et al. (2021), simply aggregating knowledge from clients with heterogeneous data can disrupt the model training and impair the model's performance on local tasks, rather than enhancing them. Recently, more studies have applied FL to various graph learning scenarios, such as node classification with distributed subgraphs Zhang et al. (2021); Chen et al. (2020); Yang et al. (2024) and distributed knowledge graph completion Chen et al. (2021b); Zhang et al. (2022). However, these techniques do not adequately address cross-client graph data heterogeneity which can widely occur in real-world scenarios.

To gain a deeper insight into the relation between data heterogeneity and FL efficacy, we conduct synthetic experiments on the molecules dataset NCI1 by simulating different levels of heterogeneity. In these experiments, all clients are partitioned into two groups, where clients within the same group share a consistent label distribution. For the homogeneous setting where two groups have the same distribution (see Figure 1(a)), the advantage of FedAvg McMahan et al. (2017) significantly expands as we distribute the data to more clients and increase the data scarcity. However, for the moderately heterogeneous setting where the difference between two groups is acceptable (see Figure 1(b)), FedAvg surpasses Self-training only when local data are extremely scarce. For the highly heterogeneous setting where cross-group divergence continues to

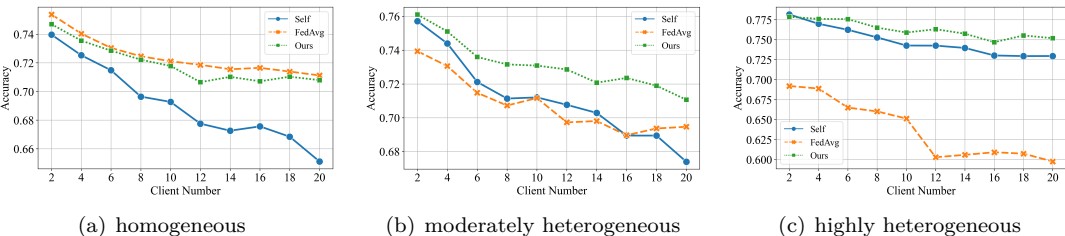

(a) homogeneous        (b) moderately heterogeneous        (c) highly heterogeneous

Figure 1: Experimental analysis on the influence of data heterogeneity over FL on a real-world graph dataset (NCI1 Morris et al. (2020)). Clients within the same group share consistent label distributions while the heterogeneity between groups grows as the setting changes from homogeneous to highly heterogeneous.

grow (see Figure 1(c)), FedAvg significantly undermines self-training instead of enhancing it. Our model performs well across all settings by properly handling different degrees of heterogeneity.

To properly address data heterogeneity among clients, personalized FL Li et al. (2021); Deng et al. (2020) has been proposed to learn client-specific models. Existing methods typically comprise two main components Chen et al. (2022a): knowledge sharing and model personalization. The first component aims to construct global models through local model aggregation, while the second one focuses on optimizing local models under the guidance of global models. Current methods primarily emphasize model personalization while overlooking the importance of effective knowledge-sharing. Such implementation may fail to recognize diverse dependencies among different clients. In a complex setting where the data distributions of clients vary from each other, uniform aggregation across all clients may result in a low-quality global model, causing the degradation of local model performance. To design a more effective knowledge-sharing mechanism, it is beneficial to reveal the latent relationships between clients. As a powerful tool for modeling the complex relationships among different entities, graphs have been used in many scenarios. Under the guidance of a relationship network with links between similar clients, we can strategically perform knowledge sharing. This approach allows clients to selectively benefit from knowledge consistent with their own, while mitigating the influence of incongruous information from disparate clients, thus enhancing the overall performance of FL.

However, the lack of supervision makes quantifying the relationship among clients challenging. For graph classification, relationships among clients are even more difficult to capture due to the complexity of graph data, which involves multi-sourced heterogeneity from node features, topological structures and the dependency between them. In this work, we propose a novel **G**raph **P**ersonalized **F**ederated **L**earning framework (a.k.a. GPFL) by integrating an unsupervised client network learning model and a GNN-based model aggregator into FL for graph classification. Specifically, in each communication round, we extract marginal parameters from local models to capture evolving client features and refer to a combination of fundamental graph properties and functional embedding to dynamically initialize the client network. A network reconstructor is then utilized to refine the client network based on the client features and the initial network. Finally, we conduct GNN-based model aggregation using the reconstructed client network. Following are the contributions of this paper.

- We design client features using marginal parameters, ensuring the capturing of evolving client characteristics along FL.

- We guide client network learning with a dynamic combination of fundamental graph properties and functional embedding. Additionally, a rule-based feature selector is tailored to identify pivotal properties across diverse domains.

- We develop a graph reconstructor that refines the client network leveraging the initial network and client features. Furthermore, our GNN-based model aggregator for FL enhances knowledge sharing, capitalizing on the underlying network relationships for improved graph classification.

## 2 Related Work

**Federated Learning with Graph** Driven by the growing research interest in Federated Learning (FL) and the advancements of Graph Neural Network (GNN) techniques, increasing work signifying the intersec-

tion of these two pivotal domains is emerging, which can be divided into two categories: graph federated learning (GFL) and federated graph learning(FGL). The former category aims to leverage the latent graph relationship among clients Chen et al. (2017; 2022a) to address the heterogeneity among clients through graph-guided knowledge sharing. In this field, the critical problem is to establish proper graph structure among clients. While Lalitha et al. (2019) follows preliminary graph structures, Caldarola et al. (2021) establishes clusters for similar clients. FGL, on the other hand, concerns the local task over graph datasets. Current FGL studies can be further categorized into inter-graph and intra-graph FGL. Works in the former category focus on graph-level local tasks such as graph classification Xie et al. (2021); Tao et al. (2022); He et al. (2021), knowledge graph completion Chen et al. (2022b), recommendation based on decentralized user-item graphs Wu et al. (2021). However, in the latter setting, clients own subgraphs of the entire global graph and there exists structural links between local graphs. So the key is to recover lost positional information through different ways, i.e., missing neighbor generation Zhang et al. (2021), link-type aware collaboration Xie et al. (2023), community discovery Baek et al. (2023a).

Though existing works have studied GFL and FGL, the intersection field where local clients own heterogeneous graph datasets remains rarely explored. The various types of heterogeneity in different graph topologies and features are the main problems for constructing a reasonable graph structure among clients. Chen et al. (2021a) designs a self-supervised FL framework to capture the heterogeneity among clients but only considers basic graph structure instead of high-order graph topology. Xie et al. (2021) analyzes the non-IID graph property over cross-domain graph datasets including cluster coefficient and shortest path length etc, but only performs knowledge sharing in cluster manner. None of them consider the unique challenge of heterogeneous graph data and addresses the challenge under the guidance of properly designed client network.

**Graph Structure Learning** Graph Neural Network (GNN) is a powerful tool for exploiting graph-structured data. However, its efficacy heavily depends on the quality of graph structure, which can be impaired by misleading links. To mitigate these limitations, Graph Structure Learning (GSL) has been proposed to optimize graph structure and representations simultaneously. For instance, GLCN Jiang et al. (2019) constructs graph structure based on node feature similarity. GAE Kipf & Welling (2016) initially embeds raw graphs into latent spaces, refining the graph structure based on embeddings.

Recently, there have been many works focusing on further enhancing the quality of graph structures through incorporating external constraints *i.e.* knowledge distillation Wu et al. (2022), graph family preliminaries Balcan & Sharma (2021); Pu et al. (2021), supervised information corresponding to certain downstream tasks Fatemi et al. (2021); Wang et al. (2021). Furthermore, for unstructured data, including images Han et al. (2022), molecules Yu & Gao (2022), GSL can also uncover the latent graph relationship. In this way, GSL facilitates the application of GNN to learn superior representations by considering the interrelationships among data samples rather than treating them separately. However, existing GFL works rely on inflexible metric-based or direct optimization methods to extract the latent client network, overlooking the potential of GSL in learning informative client relationships.

## 3 Methodology

### 3.1 Problem Formulation

Inspired by the insights gained from the synthetic experiment and existing graph-based FL works Chen et al. (2022a), we propose a novel **G**raph **P**ersonalized **F**ederated **L**earning (GPFL) framework, which is a pioneering client-network-based personalized FL framework tailored for graph applications.

Figure 2 shows an overview of GPFL. It consists of four main parts: 1. the design of client features $X_{\mathcal{C}}$ based on marginal parameters derived from local parameters to capture the evolving characteristics of each client, 2. the graph property guidance for initializing property-based client network $A_0$ 3. the client network update component for refining the property-based client network $A_0$ by performing a weighted average with a similarity network $A_E$ considering the functional embedding of each local model over a set of random graphs to generate a dynamic initial network $A_t$ for communication round $t$, and 4. the knowledge sharing component where the learned client network $\hat{A}$ is reconstructed from designed client features $X_{\mathcal{C}}$ and initial

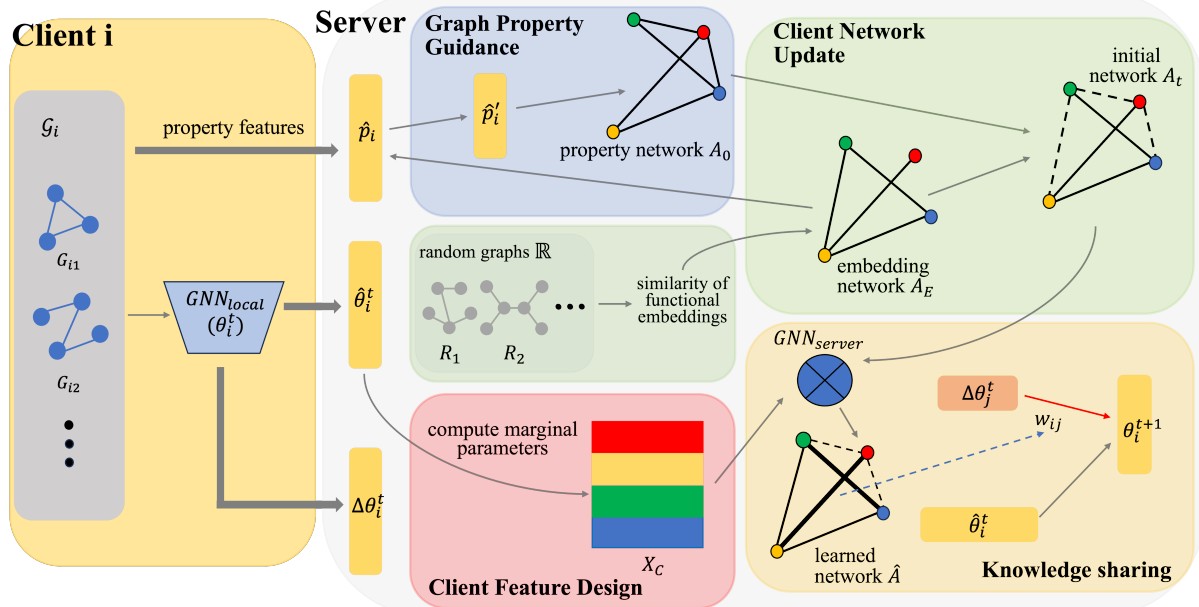

Figure 2: Overview of GPFL. Knowledge sharing is performed under the guidance of the client network, which is reconstructed from marginal parameters. The initialization of this network is guided by fundamental graph properties and functional embeddings to ensure consistent federated collaboration.

client network $A_t$ and then the weighted aggregation of local models is conducted under its guidance. The detailed pseudo-code is presented in Algorithm 1

Assume there are k clients $\{\mathcal{C}_1, \mathcal{C}_2, ..., \mathcal{C}_k\}$ with corresponding local graph datasets $\{\mathcal{G}_1, \mathcal{G}_2, ..., \mathcal{G}_k\}$. Each local dataset $\mathcal{G}_i$ contains a set of graph $\{G_{i1}, G_{i2}, ...\}$, where $G_{ij} = (V_{ij}, E_{ij}, X_{ij}, y_{ij}) \in \mathcal{G}_i$ is a graph with a node set $V_{ij}$, an edge set $E_{ij}$, a node feature set $X_{ij}$ and the graph label $y_{ij}$. For each client $\mathcal{C}_i$, its local task is to train an individual graph classification model $GNN_{local}$ parameterized by $\theta_i$:

$$\theta_i = \operatorname{argmin} \frac{1}{\|\mathcal{G}_i\|} \sum_j l(GNN_{local}[G_{ij}; \theta], y_{ij}), \tag{1}$$

where $l$ is the loss function for the local task.

During FL, at communication round $t$, each client follows Eq 1 to compute $\hat{\theta}_i^t$, which is close to the real solution for the local task. The local updates $\Delta\theta_i^t = \hat{\theta}_i^t - \theta_i^t$ are then uploaded to the server for aggregation:

$$\theta_i^{t+1} \leftarrow \theta_i^t + \underset{j \in \mathbf{N}(i)}{AGGR}(\phi(\Delta\theta_i^t, \Delta\theta_j^t, w_{ij})), \tag{2}$$

where $\theta_i^t$ and $\Delta\theta_i^t$ are the parameter and update for all client $i$. $W = [w_{ij}]$ is the knowledge-sharing weight matrix between clients in communication round $t$. $\mathbf{N}(i)$ is the neighborhood of client $i$ and $AGGR$ is the aggregation function. Our goal is to optimize the personalized FL loss $\mathcal{L}$ under iteratively gradient sharing:

$$\min_{\{\theta_i\}, W} \mathcal{L}(\{\mathcal{G}_i; \theta_i\}, W) = \mathbb{E}_i[l(\mathcal{G}_i; \theta_i)], \tag{3}$$

The knowledge-sharing matrix W is gained by combining the normalized client graph $\hat{A}$ and the identity matrix I together: and summing it with the identity matrix $I$,

$$W = \gamma I + (1 - \gamma) Normalize(\hat{A}), \tag{4}$$

where $\gamma$ is the hyperparameter used to maintain a balance between accepting external knowledge and fitting local data. The discrete client network $\hat{A}$ is constructed on the server with a simple but effective structure

---

**Algorithm 1** Graph Personalized Federated Learning

---

**Inputs:**
Clients $\{\mathcal{C}_1, \mathcal{C}_2, ..., \mathcal{C}_k\}$, Local datasets $\{\mathcal{G}_1, \mathcal{G}_2, ..., \mathcal{G}_k\}$, Initial local models $\{\theta_1^0, ..., \theta_k^0\}$
Random graph set $\mathbb{R}$, Initial client network $A_0$, $N$ iterations for training $GNN_{server}$
**Outputs:**
Final local models $\{\theta_1^T, ..., \theta_k^T\}$
1: **for** communication round t = 0,1,...,T-1 **do**
2:     **for** each client $\mathcal{C}_i$ **do**
3:         $\hat{\theta}_i^t \leftarrow$ Local Update$(\mathcal{G}_i; \theta_i^t)$, $E_{\mathcal{C}_i} \leftarrow GNN_{local}(\mathbb{R}; \hat{\theta}_i^t)$
4:     **end for**
5:     $X_{\mathcal{C}_i} \leftarrow Flatten(\hat{\theta}_i^t - \frac{1}{k}(\sum_{j=1}^k \hat{\theta}_j^t))$
6:     $A_E \leftarrow E_{\mathcal{C}} E_{\mathcal{C}}^T$, $A_t \leftarrow \beta * A_{t-1} + (1-\beta) * A_E$
7:     **for** i = 0,1,...,N-1 **do**
8:         $Z_{\mathcal{C}} \leftarrow GNN_{server}(X_{\mathcal{C}}, A_t)$, $\hat{A} \leftarrow \mathbb{I}(softmax(Z_{\mathcal{C}} Z_C^T) \geq h)$
9:         $L_{server} \leftarrow \mathbb{E}_{i,j}[H(\hat{a}_{ij})] - \mathbb{E}_{i,j}[\log(1 - \hat{a}_{ij})]$
10:        Update$(GNN_{server}, L_{server})$
11:     **end for**
12:     $W = \gamma I + (1-\gamma)Normalize(\hat{A})$
13:     **for** each client $\mathcal{C}_i$ **do**
14:         $\theta_i^{t+1} \leftarrow \theta_i^t + AGGR_{j \in \mathbf{N}(i)}(\phi(\Delta\theta_i^t, \Delta\theta_j^t, w_{ij}))$
15:     **end for**
16: **end for**

---

learning model graph auto-encoder (GAE) Kipf & Welling (2016) denoted by $GNN_{server}$. To establish $\hat{A}$, we first compute $Z_{\mathcal{C}}$, the latent embedding of clients, using $GNN_{server}$ based on client features and the input client network:

$$Z_{\mathcal{C}} = GNN_{server}(X_{\mathcal{C}}, A_t), \tag{5}$$

where $X_{\mathcal{C}}$ is the features needed for client network learning and $A_t$ is the input network starts from the initial client network $A_0$ and updated through $t$ communication rounds. Then discrete client network $\hat{A}$ is derived by filtering the inner product of $Z_{\mathcal{C}}$,

$$\hat{A} = \mathbb{I}(softmax(Z_{\mathcal{C}} Z_C^T) \geq h) \tag{6}$$

Here the $softmax$ and threshold $h$ are employed to cut out weak connections, which can assist in improving the efficiency of knowledge sharing. Theoretical analysis regarding aggregation efficiency in Section 3.4

In the following sections, we will specify how to obtain client features $X_{\mathcal{C}}$, the initial client network $A_0$, the updated input client network $A_t$, and the training process of $GNN_{server}$

### 3.2 Client Network Learning

To learn an informative client network, we propose to design client features $X_{\mathcal{C}}$ revealing the essence of local tasks and guide the learning process by constructing a property-based client network $A_0$. In communication round $t$, we first combine the property guidance with functional embedding by mixing last round initial client network $A_{t-1}$ with the similarity network $A_E$ based on local model embeddings over random graphs to get initial client network $A_t$ and then feed $X_{\mathcal{C}}$ and $A_t$ into a two-layer graph auto-encoder $GNN_{server}$ to get the final network $\hat{A}$ for knowledge sharing.

**Client feature design.** The local model parameter is a common choice for client features Chen & Zhang (2022) for its abundant information about the client. However, simply using model parameters may not be effective in FL scenarios with heavy data heterogeneity due to the similar natures of all graph classification tasks You et al. (2020). To study the impact of taking vanilla local parameters as client features under highly heterogeneous settings, we design an experiment following the same setting in Figure 1(c) where clients are divided into two groups and across-group clients exhibit significantly heterogeneous distributions compared to intra-group clients. As Figure 3 shows, even in heavily heterogeneous scenarios, the local parameters are

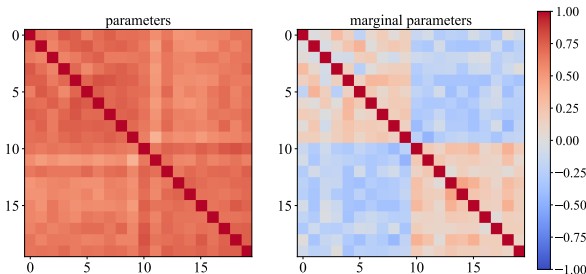

Figure 3: Similarity of local parameters and marginal parameters under high heterogeneity.

still too similar to be considered as client features due to the inherent similarities of their local tasks. As a result, constructing client relationships with vanilla local parameters is not promising in heterogeneous settings.

To address this problem, Xie et al. (2021) uses local gradients instead to construct client clusters (a particular kind of graph relationship). However, when it shows better results in a heterogeneous setting, the gradient keeps fluctuating and can not give stable results. To circumvent the aforementioned issues related to vanilla parameters and gradients, we employ the difference between local parameters and the uniform aggregated global model, termed marginal parameters, for instructing the construction of the dynamic client network,

$$X_{\mathcal{C}_i} = Flatten(\hat{\theta}_i^t - \overline{\theta}) \quad where \quad \overline{\theta} = \frac{1}{k}(\sum_{i=1}^{k} \hat{\theta}_i^t). \tag{7}$$

Here, $X_{\mathcal{C}_i}$ refers to the feature of client $i$. This approach diverges from typical normalization as it maintains the scale of each parameter that is closely related to the importance of the parameters. As shown in Figure 3, the heatmap of marginal parameters, compared with raw parameters, shows a clear pattern indicating the group relationship among clients, i.e., the similarity of clients within groups is high while across groups is low. The marginal parameters, as client features, are not only more expressive than vanilla parameters but also more stable than gradients since their convergence is ensured by the convergence of local models, which makes it beneficial for learning a high-quality client network.

**Client network update.** As local models keep evolving across the communication rounds, always starting with the property-based network $A_0$ may not be adequate, regardless of how sophisticated the initialization strategy is or even with the most powerful graph encoders, since $A_0$ cannot track the dynamics of evolving local models along FL. To resolve this issue, we designed a client network updating mechanism to ensure that the client network is up-to-date for feeding into the graph learner. Since we have already taken the local parameters as input features, we consider the functional embedding here since it provides additional perspective on the local model. To achieve this, we first generate random graphs using Erdős–Rényi methods (Paul (1959); Gilbert (1959)) as a global dataset $\mathbb{R}$. We then obtain the functional embedding of each client by aggregating the embedding of each graph in $\mathbb{R}$ and compute the cosine similarity between each pair of functional embeddings to form an embedding similarity network $A_E$. In every communication round $t$ ($t > 0$), we update the initial client network $A_t$ with the following equation:

$$A_t = \beta * A_{t-1} + (1 - \beta) * A_E, \tag{8}$$

where $\beta$ is the hyperparameter used for maintaining a balance between preserving the existing relationship between clients constructed in the last communication round and adjusting to a new relationship given by the embedding similarity network. By incrementally incorporating functional embedding into the initial property guidance, we ensure the dynamic capture of the inter-client relationships.

**Client network reconstruction.** To combine client features $X_{\mathcal{C}}$ and the updated client network $A_t$, we train a graph autoencoder(GAE) Kipf & Welling (2016), denoted as $GNN_{server}$ to reconstruct $A_t$, aiming at eliminating unnecessary client connections while amplifying the critical ones. As in the FL setting, it is challenging to obtain supervised information at the server without compromising local data privacy. We train $GNN_{server}$ in unsupervised manner by $N$ iterations on each communication round, leveraging the

regularization term from Ying et al. (2019):

$$L_{server} = \mathbb{E}_{i,j}[H(\hat{a}_{ij})] - \mathbb{E}_{i,j}[\log(1 - \hat{a}_{ij})], \tag{9}$$

where $\hat{a}_{ij} = sigmoid(Z_{\mathcal{C}:i}Z_{\mathcal{C}:j}^T)$ represents the probability that client $i$ and client $j$ should be connected and $H$ is the entropy function. The first expectation penalizes for uncertain edge, i.e., $\hat{a}_{ij}$ around 0.5, while the second one stresses the sparsity of the client network. As $X_{\mathcal{C}}$ and $A_t$ have captured the central information of each client, $GNN_{server}$ would be able to eliminate disruptive links and construct a more coherent collaboration network.

### 3.3 Graph Property Guidance

While our client feature design and client network updating mechanism can effectively excavate the inter-connections among clients and dynamically construct expressive client relationship networks even based on a random initial network (uniformly sample each entry from $[0, 1]$), relying solely on these two components may be suboptimal for overlooking the prior topology knowledge for clients. Therefore, to incorporate the guidance of fundamental graph properties into the learning process, we proposed a property-guided initialization that can leverage the inherent topological characteristics of clients, thus laying a more solid foundation for client relationship capture.

We first search for a set $\mathbb{P}$ of fundamental graph properties, including network entropy Xu et al. (2023), density, average degree, degree variance, scale-free exponent, and average closeness centrality. These graph properties are computationally straightforward yet strong in expressing essential information about the corresponding graph samples. Specifically, network entropy, defined through random walk Cover (1999); Burda et al. (2009); Spitzer (2013), is a potent way to measure the information quantity of the graph, as a special type of entropy rate and an important property for distributions. The network entropy can be efficiently derived in the case that the graph is connected (ubiquitous in real-world scenarios) with the formula,

$$p_{entropy} = \frac{1}{|E|} \sum_i d_i \log d_i. \tag{10}$$

The four features, density, average degree, degree variance, and scale-free exponent, theoretically proved closely related to network entropy in Xu et al. (2023), are important features regarding the degree distribution which is a key component of graph topology. Apart from them, We also use the average closeness centrality, i.e., average shortest distance among vertices, for a high-order measurement of graph topology.

Then we construct the initial client network with the identified property set $\mathbb{P}$. Each property is viewed as a function $P : G \to v$, that embeds graph samples with a single scalar. Therefore, by incorporating all properties, the intrinsic topological characteristics of each graph $G$ can be demonstrated with a feature vector $p_G = (P(G))P \in \mathbb{P}$, which can be further aggregated to form the client-level property feature $p_{\mathcal{C}_i}$ for client $\mathcal{C}_i$. Finally, the initial client network based on property set $\mathbb{P}$ is constructed with cosine similarity, $\boldsymbol{A}_{\mathbb{P}} = \{cos(p_{\mathcal{C}_i}, p_{\mathcal{C}_j})\}$.

However, as a single property can hold different significance for graphs from different domains due to their varying nature, indiscriminately using all properties in all cases may obscure the key information. Thus, we further introduce a rule-based feature selector to select core properties $\mathbb{P}^*$ specific to the current domain from all candidates $\mathbb{P}$ by iteratively removing insignificant properties. Firstly, we construct the initial client network $\boldsymbol{A}_{\mathbb{P}_0}(\mathbb{P}_0 = \mathbb{P})$ with all properties following the process stated above. After that, we persistently manipulate one of the properties to minimize the gap between $A_{\mathbb{P}_0}$ and functional embedding network $\boldsymbol{A}_E$, enabling us to select pertinent properties tailored to the current scenario, which is described as follows:

$$P' = argmin(\|\boldsymbol{A}_{\mathbb{P}_k \setminus \{P'\}} - \boldsymbol{A}_E\|_2) \tag{11}$$

$$\mathbb{P}_{k+1} = \mathbb{P}_k \setminus \{P'\}. \tag{12}$$

The selection process ends when further removal cannot reduce the distance, or a minimal set of properties is reached. Then we construct a property-based client network $\boldsymbol{A}_0$ with the selected property set $\mathbb{P}^* = \mathbb{P}_k$, ensuring a solid starting point for subsequent client network updates and GNN-based reconstructions.

### 3.4 Aggregation Efficiency Analysis

To further improve the efficiency of aggregation (formalized in Eq(2)), as shown in Eq(5), we consider discretizing our client network with threshold $h$ instead of directly using the dense network. Theoretical analysis regarding the efficiency of aggregation for both dense and discrete client networks is as follows.

Assume that for each client $i(i = 1, 2, ..., k)$, its local model $\theta_i$ contains $F$ parameters and its degree is $d_i$ (the number of $j$ such that $i \neq j$ and client $i$ and client $j$ are linked within the client network). The time cost of addition and multiplication in Eq(2) are $T_a$ and $T_m$, respectively.

For aggregation with a dense client network, the case where all $d_i = k - 1(i = 1, 2, ..., k)$, we would naturally have a dense weight matrix $W$. Consequently, to update local parameters, each client $i$ has to first compute weighted gradients $w_{ij}\Delta\theta_j^t$ from all clients $j(j = 1, 2, ..., k)$, sum them up, which takes $kF$ multiplication and $(k-1)F$ addition, and then combine the aggregated gradients $\sum_{j=1}^k w_{ij}\Delta\theta_j^t$ with local parameters $\theta_i^t$, which needs another $F$ addition. So in this case, the aggregation of each client would cost $kF$ multiplication and addition. Since we have $k$ clients, the overall time cost is:

$$T_{dense} = k^2 F(T_a + T_m). \tag{13}$$

Then considering a discrete client network $\hat{A}$. Note that the mixup in Eq(4) would not introduce any additional link between clients since adding an identity matrix only strengthens the self-loop of each client. So the weight matrix $W$ resulting from our discrete client network $\hat{A}$ can be viewed as a sparse matrix, where each client $i$ only aggregates gradients from its neighbors in the client network and itself. In analogy to the dense case, we would have the aggregation cost of each client is $(d_i + 1)F$ addition and multiplication, where the extra $F$ addition and multiplication are used to aggregate the gradient of itself. So we have:

$$
\begin{aligned}
T_{discrete} &= \sum_{i=1}^{k}(d_i + 1)F(T_a + T_m) \\
&= (2M + k)F(T_a + T_m),
\end{aligned}
\tag{14}
$$

where $M = \frac{1}{2}\sum_{i=1}^k d_i$ is the number of edges between different clients in $\hat{A}$. And the dense network is a special case where $M = \frac{1}{2}k(k-1)$. So we have $T_{dense}/T_{sparse} = k^2/(2M + k)$. Empirically, we have $k^2/(2M + k) \approx 1.7$ through our experiments. It suggests that discretization can improve the aggregation efficiency by 70%.

### 3.5 Overall Complexity Analysis

We analyze the computational complexity of each component in GPFL per communication round. Following Sec3.4, $k$ denotes the number of clients, $F$ the number of parameters in each local model, and $E$ the dimension of functional embedding. The $GNN_{server}$ is a 2-layer GAE trained for $N$ iterations, with hidden dimensions $d_0, d_1$. The size of the random graph set $\mathbb{R}$ and $E, N, d_0, d_1$ are fixed constants across all settings.

**Client feature design.** As shown in Eq7, the complexity of calculating marginal parameters is $O(kF)$.

**Client network update.** As the size of $\mathbb{R}$ and embedding dimension $E$ are constants, obtain the functional embedding would takes $O(F)$ for each client and $O(kF)$ for all $k$ clients. Then, calculate the similarity matrix $A_E$ and update dense client network $A_{t-1}$ to $A_t$ cost $O(k^2)$, resulting in an overall complexity of $O(kF+k^2)$.

**Client network reconstruction.** The process of encoding $X_{\mathcal{C}}$ and $A_t$ into latent embedding $Z_{\mathcal{C}}$ can be formalized as:

$$Z_{\mathcal{C}} = A_t\sigma(A_t X_{\mathcal{C}} W_0)W1, \tag{15}$$

where $W_0 \in R^{F \times d_0}$ and $W_1 \in R^{F \times d_0}$. In the first layer, $X_{\mathcal{C}}$ is first mapped to $d_0$ dimension, which costs $O(kF)$ since $d_0$ is a constant. Then, aggregation through $A_t$ takes $O(k^2)$, yielding a total complexity of $O(k^2 + kF)$. Similarly, the second layer costs $O(kd_0 + k^2) = O(k^2)$. Then the decode process, as shown in Eq 6, takes $O(k^2)$. Consequently, the time cost for reconstructing $A_t$ is $O(k^2 + kF)$. After that, calculate the reconstruction loss in Eq 9 takes $O(k^2)$. The above process is repeated for $N$ iterations each communication round. Since $N$ is a constant, the overall complexity should be $O(k^2 + kF)$.

**Aggregation.** As demonstrated in Sec 3.4, considering that $M \geq k$, the complexity is $O(MF)$.

Combining all components, the overall time cost should be $O(k^2 + MF)$. As local model size $F$ typically dominates the number of clients $k$, the aggregation becomes the primary computation bottleneck. Such observation underscores the necessity of our aggregation optimization in Sec 3.4.

### 3.6 Discuss on GPFL

As we focus on constructing the client network in a self-supervision manner, only with structural prior formalized by regularization terms, the local data are not sufficiently exploited. Meanwhile, privacy and efficiency problems are not extensively explored in this work. Detailed discussions on the limitations of this work concerning these three problems are listed below.

**Semi-Surpervised Learning manner for refining client network** One limitation of our model is that we need a hyperparameter $\alpha$ to balance between adopting external gradients from other clients or believing in the knowledge contained in local gradients. Fortunately, Zhang et al. (2023) provides a feasible way to learn client-specific $\alpha_i$ in a supervised manner, where we can construct a local model parametrized by $\alpha$ in the following formula:

$$\theta_i^{t+1} = \theta_i^t + (1 - \alpha_i)\Delta\theta_i^t + \alpha_i \cdot (\sum_{j=1}^{k} w_{ij}\Delta\theta_j^t), \tag{16}$$

where $\theta_i^t$ and $\Delta\theta_j^t$ have the same definition in Section 3.1. Then we can train these local models parametrized by $\alpha_i$ over local datasets and iteratively optimize $\alpha_i$ with gradient descent over local loss functions to reach an ideal result. Although simply adding this component will cause a performance drop rather than an improvement, let alone significant time costs for this second-round local training towards the convergence of $\alpha_i$ within a single communication round, it's still a promising way to further refine the client network in a semi-supervised learning manner which can be explored in future work.

**Data Privacy** In our setting of FL over graphs, the global graph reconstruction task does not require any type of inter-client or client-server data transmission. For the functional embedding step, instead of building a global dataset by sampling representative local graphs, we employ random graphs, which shield our GPFL from the data leakage problem. However, we do not consider how to deal with malicious attacks in this work. In the future, we can introduce advanced privacy protection techniques to ensure the safety.

**Efficiency** In this work, we focus less on the efficient implementation of our framework, and the current training process takes longer time than some baseline methods we compare e.g. FedAvg, GCFL to ensure the convergence of the client network, although as shown in Figure 6, the time cost of constructing the client network is almost independent of the scale of local datasets and will become less significant as the scale continues to grow. In the future, we could further improve the efficiency of GPFL by adopting the early-stop strategy in the learning process and only performing adaptive learning over clients with uncertain relationships in each communication round.

## 4 Experiments

### 4.1 Experimental Settings

**Datasets.** We utilize the three most widely used graph classification benchmark datasets from two domains Morris et al. (2020), including two molecules datasets (NCI1, Yeast) and a bioinformatics dataset (PRO-

TEINS). Among them, NCI1 and PROTEINS are relatively small, containing about four and one thousand graphs separately, while Yeast is a large dataset containing nearly 80,000 graphs. However, considering the severe label distribution skew existing in the raw Yeast dataset, we conduct downsampling to create a uniform subset containing 20,000 graphs. This approach is taken because designing effective split methods to control heterogeneity becomes challenging when the raw dataset is already lopsided. The data details are as follows (Table 1).

Table 1: The statistics of datasets.

| Dataset | Statistics | | | |
|---------|---------|---------|---------|---------|
| | ♯graphs | avg. ♯nodes | avg. ♯edges | ♯ classes |
| NCI1 | 4110 | 29.87 | 32.30 | 2 |
| Yeast | 79601 | 21.54 | 22.84 | 2 |
| PROTEINS | 1113 | 39.06 | 72.82 | 2 |
| Yeast (sampled) | 19136 | 22.77 | 24.22 | 2 |

**Research questions.** To comprehensively evaluate the effectiveness and contribution of our proposed framework, we formulate four research questions as follows that will guide our empirical investigations:

- **RQ1**: How does GPFL compare to other widely adopted FL frameworks in graph classification tasks?

- **RQ2**: How do the proposed client network learning framework and network initialization mechanisms individually contribute to the overall performance?

- **RQ3**: What does the inferred client network reveal, and how do the networks learned with different combinations of components differ from each other?

- **RQ4**: How do the proposed client network learning framework and graph initialization mechanisms individually contribute to the overall time cost?

We design label heterogeneity settings following a practical data split mechanism Wang et al. (2020); Lee et al. (2021); Luo et al. (2021), which is controlled by the Dirichlet distribution $Dir(\alpha)$. The setting becomes more heterogeneous as the value of $\alpha$ decreases. We consider $\alpha = 0.5, 1, 5$ to represent strong heterogeneity, moderate heterogeneity, and weak heterogeneity in real-world scenarios, respectively. These settings are combined with varying levels of data scarcity, represented by client numbers k = 15, 20, and 25, yielding a total of nine distinct combinations.

**Compared methods.** We employ **self-train** as the initial baseline to assess whether FL can enhance client performance. In this baseline, each client starts from the same initial model downloaded from the server and conducts independent local training without collaboration. For FL baselines, we employ basic FL algorithms **FedAvg** McMahan et al. (2017), **FedProx** Li et al. (2020), and five widely adopted personalized FL algorithms. Among those personalized methods, **SCAFFOLD** Karimireddy et al. (2020) is based on gradient adjustments, **GCFL** Xie et al. (2021) is based on clustering, **FedStar** Tan et al. (2023) adopts a decoupled sharing strategy, **FED-PUB** Baek et al. (2023b) and **FedSelect** Tamirisa et al. (2024) construct masks to perform selective parameter sharing, and **FedAMP** Huang et al. (2021) and **pFedGraph** Ye et al. (2023) also follow a graph-based aggregation manner. For the graph classification model, we employ **GIN** (Graph Isomorphism Network) Xu et al. (2019), a simple yet powerful GNN for graph-level tasks. We fix the architecture and hyper-parameters of the local model for all baselines in all experiment settings to control the experiment.

**Evaluation metrics.** We measure the performance of different FL algorithms using the average accuracy of all local models evaluated on the local test datasets and report the accuracy achieved in the last communication rounds as a reference for all methods. All experiments are run with a five-fold cross-validation for three repetitions under fixed random seed 0.

Table 2: Average accuracy on NCI1, Yeast, and PROTEINS datasets under multiple clients and different values of $\alpha$. $\alpha$ is the parameter of the Dirichlet distribution. The heterogeneity among clients increases when the $\alpha$ decreases.

| Dataset(♯clients) | NCI1(25) | | | NCI1(20) | | | NCI1(15) | | |
|---|---|---|---|---|---|---|---|---|---|
| skew rate | $\alpha=0.5$ | $\alpha=1$ | $\alpha=5$ | $\alpha=0.5$ | $\alpha=1$ | $\alpha=5$ | $\alpha=0.5$ | $\alpha=1$ | $\alpha=5$ |
| self-train | 0.7766(0.012) | 0.7124(0.011) | 0.6591(0.010) | 0.7925(0.008) | 0.7259(0.015) | 0.6901(0.018) | 0.8121(0.014) | 0.7261(0.014) | 0.7093(0.012) |
| FedAvg | 0.5756(0.029) | 0.6163(0.014) | 0.6474(0.025) | 0.6486(0.011) | 0.6583(0.018) | 0.6832(0.011) | 0.6339(0.019) | 0.6646(0.013) | 0.6909(0.019) |
| FedProx | 0.5754(0.022) | 0.6162(0.017) | 0.6459(0.020) | 0.6432(0.016) | 0.6607(0.020) | 0.6873(0.011) | 0.6379(0.018) | 0.6622(0.009) | 0.6935(0.021) |
| Scaffold | 0.7012(0.022) | 0.6287(0.025) | 0.6079(0.013) | 0.7632(0.013) | 0.6972(0.020) | 0.6683(0.020) | 0.7843(0.007) | 0.6929(0.017) | 0.6822(0.021) |
| GCFL | 0.7385(0.020) | 0.7278(0.019) | 0.6822(0.025) | 0.7948(0.008) | 0.7308(0.017) | 0.7142(0.013) | 0.8137(0.013) | 0.7043(0.014) | 0.7133(0.017) |
| FedStar | 0.7771(0.007) | 0.7057(0.011) | 0.6627(0.011) | 0.7798(0.013) | 0.7207(0.011) | 0.6850(0.011) | 0.7936(0.008) | 0.7144(0.014) | 0.7085(0.016) |
| pFedGraph | 0.7668(0.011) | 0.6991(0.009) | 0.6693(0.011) | 0.7931(0.005) | 0.7118(0.011) | 0.6873(0.012) | 0.7997(0.006) | 0.7090(0.004) | 0.6948(0.012) |
| FedAMP | 0.7647(0.015) | 0.7013(0.013) | 0.6555(0.015) | 0.7892(0.009) | 0.7189(0.013) | 0.6738(0.011) | 0.8118(0.010) | 0.7141(0.010) | 0.6971(0.014) |
| FED-PUB | 0.7788(0.009) | 0.7127(0.005) | 0.6582(0.006) | 0.7956(0.008) | 0.7152(0.006) | 0.6761(0.011) | 0.8139(0.007) | 0.7124(0.006) | 0.6889(0.008) |
| FedSelect | 0.7679(0.005) | 0.6811(0.005) | 0.6445(0.004) | 0.7681(0.003) | 0.6913(0.005) | 0.6552(0.006) | 0.7838(0.005) | 0.6922(0.005) | 0.6599(0.011) |
| GPFL | **0.7888(0.012)** | **0.7378(0.010)** | **0.6941(0.009)** | **0.8109(0.010)** | **0.7489(0.014)** | **0.7185(0.010)** | **0.8241(0.012)** | **0.7509(0.012)** | **0.7274(0.013)** |

| Dataset(♯clients) | Yeast(25) | | | Yeast(20) | | | Yeast(15) | | |
|---|---|---|---|---|---|---|---|---|---|
| skew rate | $\alpha=0.5$ | $\alpha=1$ | $\alpha=5$ | $\alpha=0.5$ | $\alpha=1$ | $\alpha=5$ | $\alpha=0.5$ | $\alpha=1$ | $\alpha=5$ |
| self-train | 0.7862(0.004) | 0.7171(0.007) | 0.6816(0.007) | 0.8010(0.005) | 0.7130(0.005) | 0.6780(0.006) | 0.8158(0.006) | 0.7127(0.007) | 0.6899(0.007) |
| FedAvg | 0.6245(0.009) | 0.6557(0.004) | 0.6585(0.006) | 0.6349(0.005) | 0.6527(0.008) | 0.6608(0.007) | 0.6316(0.006) | 0.6570(0.004) | 0.6615(0.008) |
| FedProx | 0.6206(0.008) | 0.6545(0.005) | 0.6586(0.006) | 0.6326(0.006) | 0.6525(0.009) | 0.6604(0.008) | 0.6284(0.007) | 0.6575(0.004) | 0.6614(0.007) |
| Scaffold | 0.7803(0.006) | 0.7154(0.006) | 0.6744(0.009) | 0.7909(0.004) | 0.7046(0.006) | 0.6735(0.007) | 0.8078(0.006) | 0.7065(0.005) | 0.6796(0.007) |
| GCFL | 0.7767(0.010) | 0.7264(0.004) | 0.6869(0.006) | 0.7985(0.005) | 0.7133(0.006) | 0.6835(0.008) | 0.8157(0.006) | 0.7178(0.007) | 0.6937(0.008) |
| FedStar | 0.7722(0.004) | 0.7020(0.006) | 0.6532(0.006) | 0.7810(0.006) | 0.6887(0.005) | 0.6551(0.008) | 0.8064(0.006) | 0.6915(0.005) | 0.6725(0.006) |
| pFedGraph | 0.7806(0.004) | 0.7213(0.003) | 0.6845(0.007) | 0.7933(0.003) | 0.7072(0.006) | 0.6782(0.006) | 0.8114(0.006) | 0.7130(0.004) | 0.6879(0.006) |
| FedAMP | 0.7837(0.005) | 0.7184(0.006) | 0.6761(0.008) | 0.7941(0.004) | 0.7023(0.006) | 0.6710(0.006) | 0.8126(0.007) | 0.7115(0.005) | 0.6844(0.004) |
| FED-PUB | 0.7673(0.002) | 0.6921(0.004) | 0.6396(0.004) | 0.7836(0.003) | 0.6796(0.002) | 0.6406(0.003) | 0.7977(0.002) | 0.6974(0.003) | 0.6440(0.004) |
| FedSelect | 0.7605(0.002) | 0.6798(0.003) | 0.6252(0.002) | 0.7667(0.001) | 0.6757(0.002) | 0.6381(0.002) | 0.7874(0.001) | 0.6764(0.003) | 0.6488(0.002) |
| GPFL | **0.7911(0.004)** | **0.7272(0.005)** | **0.6914(0.006)** | **0.8020(0.003)** | **0.7210(0.005)** | **0.6887(0.007)** | **0.8203(0.004)** | **0.7262(0.006)** | **0.6968(0.008)** |

| Dataset(♯clients) | PROTEINS(25) | | | PROTEINS(20) | | | PROTEINS(15) | | |
|---|---|---|---|---|---|---|---|---|---|
| skew rate | $\alpha=0.5$ | $\alpha=1$ | $\alpha=5$ | $\alpha=0.5$ | $\alpha=1$ | $\alpha=5$ | $\alpha=0.5$ | $\alpha=1$ | $\alpha=5$ |
| self-train | 0.7576(0.012) | 0.7383(0.037) | 0.6883(0.026) | 0.7609(0.023) | 0.7198(0.025) | 0.7052(0.031) | 0.7927(0.016) | 0.7303(0.023) | 0.7337(0.012) |
| FedAvg | 0.7123(0.025) | 0.7062(0.047) | 0.7037(0.008) | 0.7100(0.033) | 0.7332(0.036) | 0.7172(0.019) | 0.7185(0.031) | 0.7197(0.027) | 0.7068(0.017) |
| FedProx | 0.7192(0.025) | 0.7069(0.040) | 0.7045(0.011) | 0.7043(0.033) | 0.7310(0.039) | 0.7194(0.024) | 0.7192(0.031) | 0.7247(0.030) | 0.7054(0.019) |
| Scaffold | 0.7320(0.014) | 0.7128(0.036) | 0.7116(0.012) | 0.7343(0.022) | 0.7391(0.029) | 0.7269(0.020) | 0.7390(0.022) | 0.7219(0.020) | 0.7101(0.018) |
| GCFL | 0.7450(0.035) | 0.7229(0.026) | 0.6996(0.022) | 0.6973(0.035) | 0.7382(0.020) | 0.7137(0.022) | 0.7895(0.024) | 0.7330(0.028) | 0.7353(0.019) |
| FedStar | 0.7081(0.022) | 0.6984(0.028) | 0.6591(0.031) | 0.7468(0.020) | 0.6835(0.026) | 0.6767(0.036) | 0.7481(0.013) | 0.7139(0.022) | 0.6972(0.023) |
| pFedGraph | 0.7294(0.023) | 0.6979(0.034) | 0.6940(0.026) | 0.7089(0.019) | 0.6885(0.025) | 0.6946(0.028) | 0.7567(0.017) | 0.6946(0.030) | 0.7116(0.033) |
| FedAMP | 0.7467(0.017) | 0.7342(0.019) | 0.6829(0.030) | 0.7523(0.028) | 0.7145(0.043) | 0.6904(0.020) | 0.7626(0.016) | 0.7251(0.026) | 0.7211(0.023) |
| FED-PUB | 0.7607(0.015) | 0.7389(0.006) | 0.7133(0.02) | **0.7723(0.009)** | **0.7431(0.021)** | **0.7357(0.021)** | 0.7830(0.021) | 0.7510(0.025) | 0.7330(0.024) |
| FedSelect | 0.7631(0.012) | 0.7186(0.016) | 0.6976(0.012) | 0.7499(0.016) | 0.7321(0.018) | 0.6859(0.030) | 0.7519(0.015) | 0.7315(0.012) | 0.7216(0.019) |
| GPFL | **0.7787(0.015)** | **0.7460(0.028)** | **0.7155(0.029)** | 0.7686(0.030) | 0.7401(0.036) | 0.7201(0.031) | **0.8059(0.008)** | **0.7600(0.020)** | **0.7500(0.024)** |

**Parameter settings.** We utilize three-layer GINs with a hidden size of 64 as local models. Local training uses a batch size of 128, the Adam Kingma & Ba (2014) optimizer with the learning rate of $1e^{-3}$ and the weight decay of $5e^{-4}$. All FL methods are trained for 200 communication rounds with 1 local epoch in each communication round. For GPFL, we generate 20 random graphs with 30 nodes to compute functional embedding. The graph learner is trained for 100 epochs during each communication round. And the hyperparameters $\gamma$ in Eq 4 and $\beta$ in Eq 8 are set to 0.95, 0.95 across all settings, respectively. All codes and data can be found in `https://github.com/Jiachen2cc/Graph-Personalized-Federated-Learning`.

## 4.2 Overall Performance (RQ1)

As can be observed from Table 2, our proposed framework can significantly improve the performance of local graph classification tasks. As the degree of heterogeneity intensifies (decreasing $\alpha$), the effectiveness of FedAvg drops accordingly. Specifically, in relatively homogeneous settings (large $\alpha$) with a lack of data (large $k$), FedAvg achieves comparable performance with or even holds an advantage over self-training. Conversely, in heterogeneous settings (small $\alpha$) with relatively sufficient local data (small $k$), self-train demonstrates superior performance compared to FedAvg. GPFL outperforms both approaches across these settings by considering heterogeneity and providing local clients with network-based knowledge sharing, which enables similar clients to share common knowledge while preventing them from being influenced by heterogeneous users. For all datasets with all combinations of $\alpha$ and $k$, our GPFL framework achieves $1.57\% - 8.03\%$ performance gain over self-train and FedAvg on average. While personalized FL baselines SCAFFOLD, GCFL, FedStar, FED-PUB, and FedSelect demonstrate decent performance, they cannot compete with our approach due to their inflexible knowledge-sharing mechanisms. Furthermore, even though FedAmp and pFedGraph also employ graph-based knowledge sharing, their expressiveness is limited as they rely on

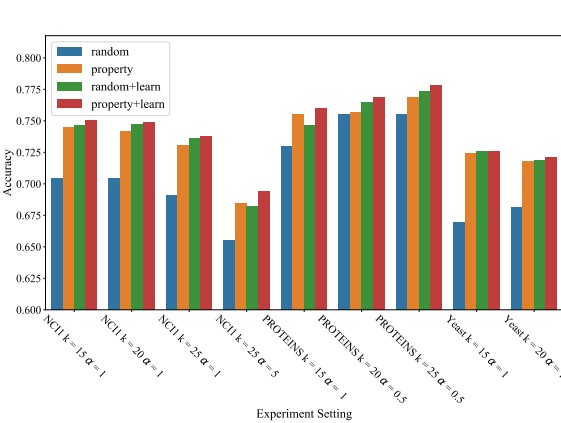

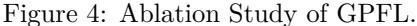

Figure 4: Ablation Study of GPFL.

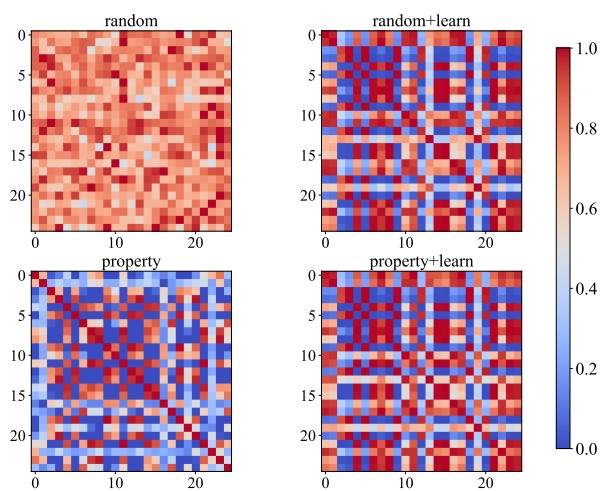

Figure 5: Client Network Visualization on NCI1 dataset with 25 clients and $\alpha = 0.5$.

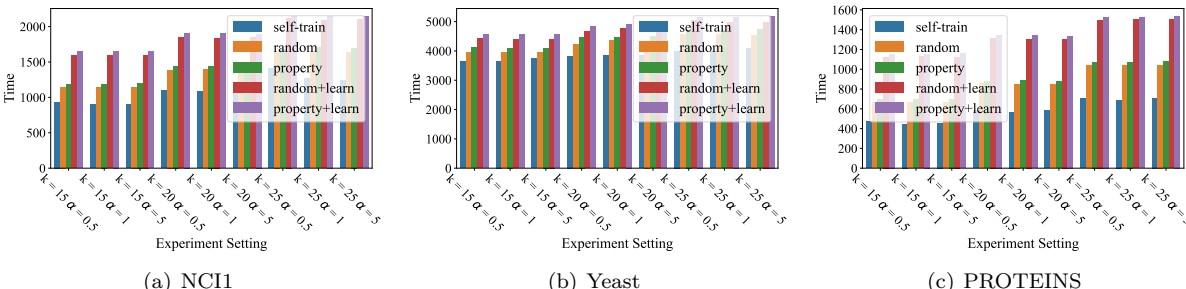

(a) NCI1      (b) Yeast      (c) PROTEINS

Figure 6: Run time analysis on property-based initialization and network learning.

vanilla parameters and their deviation from the initial parameters, which may not efficiently capture the characteristics of each client. In contrast, our approach surpasses them by explicitly modeling the client relationship as a network and applying marginal parameters and the client network to guide more precise knowledge sharing among related clients.

## 4.3 Ablation Studies (RQ2)

We conduct ablation studies to evaluate the effectiveness of each component in our model, following the same experiment setting used in Section 4.2 (see Figure 7 for complete results), as depicted in Figure 4.

**Random Initialization v.s. Property-based initialization** To evaluate the impact of graph property guidance incorporated in the initialization process, we use our GPFL framework without learning and updating; instead, we assume that the relationship between clients is fixed, either as a random (uniformly sample each weight from [0,1], denoted as "random" in Figure 4) or a property-based (denoted as "property" in Figure 4) relationship. That is, at all rounds, all clients follow the same network given in the initialization step for knowledge sharing. A random relationship network fails in all cases. The property-based initialization, capable of capturing client relationships through fundamental properties, outperforms random initialization even without adaptively graph learning; however, it is slightly inferior to GPFL.

**Fixed initialization v.s. Adaptive learning** We also examine the impact of our adaptive learning strategy including iterative network update and learning with client features. To achieve this, we evaluate the performance of adopting the learning strategy under random and property-based initialization, denoted as "random+learn" and "property+learn" separately. It can be observed that our learning strategy is effective even with random initialization as it excavates latent relationships with marginal parameters and functional

embedding. Nevertheless, learning with property-based initialization still outperforms as the model starts from a meaningful client network.

### 4.4 Client Network Visualizations (RQ3)

Figure 5 illustrates the comparison of the average client network generated using the four different methods listed in Section 4.3. Compared to the random client network, there are certain connections between the property-based client network and the learned client network, demonstrated by relative intensities. For example, in the property-based network, clients 6, 7, and 8 show strong connections with each other, a pattern that is also observed in the learned network. Given that the network learned from random initialization is derived without any property preliminaries, relying solely on local parameters and functional embedding, this observed similarity suggests that fundamental properties of graph samples can, to some extent, reflect the inherent characteristics of clients. Moreover, regardless of whether the initialization is random-based or property-based, we consistently obtain similar learned networks. This finding further demonstrates the robustness of GPFL.

### 4.5 Runtime Analysis (RQ4)

To measure the time efficiency of GPFL and the time cost of each component within it, we further conduct a runtime analysis on the three datasets with all settings. As shown in Figure 6, our property-based initialization incurs only a 5% increase in runtime compared to random initialization, as we only consider fundamental properties of graph samples that can be computed efficiently, while this slight sacrifice in runtime significantly enhances performance. And in real-world scenarios, the efficiency of property-based initialization can be further improved by precomputing and storing the property values of each graph. Furthermore, as GPFL only considers certain client properties, the time cost of learning only grows linearly with the number of clients and is almost independent of local dataset size. As demonstrated in Figure 6, while GPFL might entail heavier time costs for small-scale datasets like PROTEINS compared with vanilla self-train, its communication cost becomes acceptable when applied to larger datasets like Yeast which are more common in real scenarios.

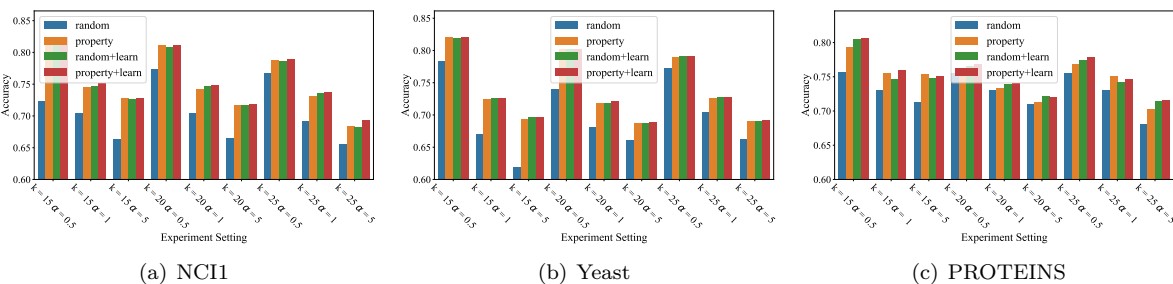

(a) NCI1        (b) Yeast        (c) PROTEINS

Figure 7: Full results of ablation studies.

## 5 Conclusion

This work focuses on FL on graph classification with local distribution heterogeneity. Specifically, we study the problem of uniform knowledge sharing in the setting where each local client owns graphs sampled from non-IID distributions. To address this problem, we propose a client-network-based personalized federated graph learning framework (GPFL) that performs GNN-based knowledge sharing based on a dynamic client network. The client network is first dynamically initialized under the guidance of fundamental properties and functional embedding and then further refined through reconstruction leveraging marginal parameters during training to ensure its effectiveness. The extensive experimental results and in-depth analysis demonstrate the effectiveness of GPFL. Moreover, we discuss the limitations of GPFL, including the lack of supervision information to further enhance the client network and the absence of optimization in terms of privacy and efficiency, which can be explored in future work.

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
