# OpenReview forum: "Graph Personalized Federated Learning via Client Network Learning"
_TMLR — Accepted by TMLR_

### Review · Reviewer_LwR7 · 2025-04-29

**Summary Of Contributions:**

This paper considers graph federated learning for graph-level classification task with heterogeneous data at the client side. The main contribution of this study is to propose a novel design of client features by using graph-guided properties and the marginal parameters. Second contribution is a client network learning method for knowledge sharing. Overall this approach seem novel to me, and the client feature design and the use of random graph model for functional embedding seem rational.

**Audience:**

Yes

**Claims And Evidence:**

Yes

**Requested Changes:**

1. Add more analysis and discussion on the structural learning model under data heterogeneity.

2. Add hyperparameter sensitivity analysis on $h$.

**Strengths And Weaknesses:**

Strengths:
1. The paper is well-written and clearly structured.
2. The paper targets a challenging problem in graph federated learning with heterogeneous data.
3. Detailed experiments are provided to support the effectiveness of the proposed framework.

Weaknesses:
1. One concern is the unsupervised way to learn the client network, since the data is heterogeneous in the client side, I am a bit concerned will the structural learning model can learn the client network effectively under this setting. This may need more justification and experiments.
2. It is not clear how to choose the hyperparameter $h$ as the threshold. It would be better to provide hyperparameter analysis, and provide guidelines for hyperparameter selection.

---

> ### Author Response · Authors · 2025-06-25
> **Reply to Reviewer LwR7**
>
> Thanks for your review, the following are our answers to your concerns.
>
> **Weakness 1.** In our setting, we choose the unsupervised graph constructor mainly to cut off noxious links between clients with inconsistent knowledge instead of mining underlying inter-client relationships. That is why we spend most effort on defining client features and the initial client network to make sure they can capture most latent links and we utilize loss functions from [1], which encourages the constructor to amplify the beneficial links while impairing the harmful ones.
>
> The unsupervised graph constructor can work well as we only need it to cut unnecessary inter-client relationships instead of exploring the complex underlying inter-client relationships.
>
> Please see the comment to all reviewers, we provide a toy experiment that shows that the learned client graph is closely related to the label distribution of each client.
>
> **Weakness 2.** Please see the comment to all reviewers for the result of sensitivity analysis.
>
> [1] Zhitao Ying, Dylan Bourgeois, Jiaxuan You, Marinka Zitnik, and Jure Leskovec. Gnnexplainer: Generating explanations for graph neural networks. In NeurIPS, 2019.

---

### Review · Reviewer_E7hw · 2025-06-07

**Summary Of Contributions:**

This paper presents a federated learning framework for graph classification tasks. The proposed framework can dynamically learn the similarities between clients based on graph structural features and GNN marginal parameters. Experimental results demonstrate the effectiveness of the proposed framework.

**Audience:**

Yes

**Broader Impact Concerns:**

NIL.

**Claims And Evidence:**

Yes

**Requested Changes:**

1. The motivation of the paper can be presented in a clear way. To me, the motivation part in the 3rd and 4th paragraphs of the Introduction is vague. The authors are suggested to use a proper example with specific explanations to show why this GPFL is proposed.
2. For related work, graph/network structure learning for generated federated learning has been investigated. Authors are suggested to survey papers published in top conferences/journals since 2023 and distinguish them from the proposed method.
3. There is no solid theoretical analysis validating the performance of the proposed GPFL.
4. More test datasets can be selected for experiments.
5. More baselines that have been published in the last two years can be selected to compare with the proposed framework.6. Model complexity should be clearly analyzed.
6. The model complexity should be clearly analyzed.

**Strengths And Weaknesses:**

1. The problem tackled in this paper is challenging in federated learning with graphs.
2. The performance of the proposed framework is promising.

---

> ### Author Response · Authors · 2025-06-25
> **Reply to Reviewer E7hw**
>
> Thanks for your review, the following are our answers to your concerns.
>
> **Requested Change 1.** Thanks for your reminder, we will polish the motivation part accordingly.
>
> **Requested Change 2.** To the best of our knowledge, there are new frameworks like ([1],[2]) utilizing client topology in PFL but none of them employ structure learning method to construct client network.
>
> **Requested Change 3.** Sorry for the missing of theoretical analysis. Though our work is not theoretical, we conduct detailed empirical analysis over the performance of GPFL, which proves it is an efficient and reliable framework for graph classification under PFL settings.
>
> Please see the comment to all reviewers for another toy experiment demonstrates that the learned client graph is closely related to the label distribution of each client.
>
> **Requested Change 4.** These three datasets are the common choices for graph classification tasks. We choose Yeast as a representative of large-scale graph datasets. Apart from them, many graph classification datasets are too small to be split into sub-datasets under the setting of $\sharp client = 25$ and $\alpha = 0.5$, where some of them would only contain graph samples from 1 class. Besides, many large-scale graph datasets possess extremely unbalanced class distribution, which makes the evaluation unfair. In this case, we think the three graph datasets we choose are enough to fairly evaluate the performance of GPFL.
>
> **Requested Change 5.**
> |Setting|Method|Performance|Method|Performance|Method|Performance|
> |------|--------|----------|------|----------|------|----------|
> |NCI1($\alpha = 0.5, \sharp clients = 25$)|GPFL|0.7888(0.012)|FedPub|0.7788(0.009)|FedSelect|0.7681(0.005)|
> |Yeast($\alpha = 0.5, \sharp clients = 25$)|GPFL|0.7911(0.004)|FedPub|0.7673(0.002)|FedSelect|0.7605(0.002)|
> |PROTEINS($\alpha = 0.5, \sharp clients = 25$)|GPFL|0.7787(0.015)|FedPub|0.7607(0.015)|FedSelect|0.7631(0.012)|
>
> We have added two new baselines FedPub([3]), FedSelect([4]). Among them, FedPub is originally designed for subgraph PFL but its key ideas can be seamlessly transformed into our setting and FedSelect is a universal PFL framework performing selective parameter aggregation
>
>
> Full results will be included in the paper.
>
> **Requested Change 6.** Thanks for your reminder, we will add a subsection at the end of Sec 3. to analyze the overall complexity.
>
> [1] Mengmeng Ma, Tang Li, and Xi Peng. Beyond the federation: Topology-aware federated learning for generalization to unseen clients. In International Conference on Machine Learning, pp. 33794–33810. PMLR, 2024.
>
> [2] Chunxu Zhang, Guodong Long, Tianyi Zhou, Zijian Zhang, Peng Yan, and Bo Yang. Gpfedrec: Graph-guided personalization for federated recommendation. In Proceedings of the 30th ACM SIGKDD Conference on Knowledge Discovery and Data Mining, pp. 4131–4142, 2024.
>
> [3] Jinheon Baek, Wonyong Jeong, Jiongdao Jin, Jaehong Yoon, and Sung Ju Hwang. Personalized subgraph federated learning. In International conference on machine learning, pp. 1396–1415. PMLR, 2023.
>
> [4] Rishub Tamirisa, Chulin Xie, Wenxuan Bao, Andy Zhou, Ron Arel, and Aviv Shamsian. Fedselect: Personalized federated learning with customized selection of parameters for fine-tuning. In Proceedings of the IEEE/CVF Conference on Computer Vision and Pattern Recognition, pp. 23985–23994, 2024.

---

### Review · Reviewer_dbah · 2025-06-11

**Summary Of Contributions:**

This manuscript addresses the challenge of graph classification in settings where high-quality labeled graph data is scarce. This manuscript proposes a novel method inspired by graph structure learning that dynamically constructs a client network based on graph data similarity. This network guides selective model sharing among clients. The approach leverages the marginal parameters of local GNNs and incorporates fundamental graph properties in its design. Experimental results on three real-world datasets validate the effectiveness and mutual reinforcement of the proposed modules.

**Audience:**

Yes

**Claims And Evidence:**

Yes

**Requested Changes:**

Please refer to the "weaknesses" part.

**Strengths And Weaknesses:**

**Pros**

- The manuscript models client features using marginal parameters to capture evolving client characteristics throughout the federated learning process.
- It leverages fundamental graph properties and functional embeddings for client network learning and introduces a graph reconstructor to enhance the network structure.
- Comprehensive evaluations are conducted to validate the effectiveness of the proposed method.

**Cons**

- As stated in the Introduction, the authors claim to design a graph reconstructor to refine the client network. However, subsequent sections lack any detailed description of this component. The authors should provide a comprehensive explanation of the graph reconstructor.
- Figure 2 fails to clearly illustrate the logical relationship between client feature design and knowledge sharing. It is unclear how the various components interact to enable personalized model learning. The authors are encouraged to include a pseudo-algorithm to clarify the training process.
- In Equation (6), the hyperparameter $h$ is used to prune weak connections. However, no sensitivity analysis is provided to assess its impact on performance. Additionally, the influence of the number of clients on the method's performance is not evaluated.
- In Table 2, the authors use FedAvg and FedProx as baselines. However, these traditional aggregation methods are designed for global model learning. The manuscript does not evaluate their performance in the context of personalized client model learning, which should be addressed.

---

> ### Author Response · Authors · 2025-06-25
> **Reply to Reviewer dbah**
>
> Thanks for your review, the following are our answers to your concerns.
>
> **Weakness 1.** The graph construtor we employed is a vanilla GAE whose input, output and loss functions are formularized by Eq 5-7. We will add another subsection under Sec 3 to define it more clearly.
>
> **Weakness 2.** Thanks for your reminder, we will include pseudo-algorithm by Figure 2 in the final version.
>
> **Weakness 3.** Please see the comment to all reviewers for the result of sensitivity analysis.
>
> **Weakness 4.** We have included model performance on 3 different client numbers(15,20,25) in Table 2.
>
> **Weakness 5.** FedAvg and FedProx are included only to showcase that traditional global model learning paradigm is not promising under PFL setting. Since many other PFL methods are already evaluated, testing the finetuned FedAvg/FedProx may not be necessary.

---

### Review · Reviewer_Qgqt · 2025-06-19

**Summary Of Contributions:**

This paper proposes GPFL, a novel framework for personalized federated learning (FL) on graph data. Key contributions include:

1. Dynamic Client Network Learning: Introduces an unsupervised method to learn a client similarity network using (a) marginal parameters (local-global model deviations) as client features, and (b) fundamental graph properties (e.g., entropy, density) for network initialization.

2. GNN-Based Aggregation: Leverages the client network to guide weighted model aggregation via a graph auto-encoder (GAE), enabling personalized knowledge sharing.

3. Heterogeneity Handling: Addresses non-IID graph data by adaptively linking clients with Graph Property Guidance, outperforming FedAvg and clustering-based methods in highly heterogeneous settings.

4. Comprehensive Evaluation: Demonstrates SOTA results on three graph datasets (NCI1, Yeast, PROTEINS) under varying heterogeneity levels (Dirichlet-α) and client counts. Ablations validate the utility of each component.

**Audience:**

Yes

**Claims And Evidence:**

Yes

**Requested Changes:**

1. The phrase *"To further improve the efficiency of aggregation (formalized in Eq. (2))"* is misleading, as Eq. (2) does not pertain to the message aggregation process.

2. The term *"marginal parameters"* appears without explanation until Eq. (8) on page 5. Consider introducing and defining this concept earlier to avoid reader confusion.

3. Although the paper analyzes aggregation efficiency theoretically, it lacks empirical measurements. Discretization operations often produce non-contiguous memory accesses, which can impair tensor operation performance. Please include experiments or benchmarks to verify and quantify this effect.

**Strengths And Weaknesses:**

**Strengths**

1. Problem Significance: Tackles challenges in federated graph learning—personalization under graph-structured non-IID data—with clear motivation from synthetic experiments (Figure 1).

2. Innovative Methodology: Marginal parameters effectively capture evolving client characteristics (Figure 3), resolving limitations of vanilla parameters/gradients; Graph property-guided initialization (e.g., entropy, centrality) provides a principled prior for client relationships; Functional embeddings from random graphs enable dynamic network updates without compromising privacy.

3. Rigorous Evaluation: Extensive experiments across 27 settings (datasets × clients × heterogeneity) show consistent gains (1.57–8.03% over baselines). Ablations (Figure 4) and visualizations (Figure 5) reinforce design choices.

**Weaknesses**
- Hyperparameter Sensitivity: Critical hyperparameters (e.g., γ, β, λ) are fixed across all experiments. Sensitivity analysis is missing.

---

> ### Author Response · Authors · 2025-06-25
> **Relply to Reviewer Qgqt**
>
> Thanks for your review, the following are our answers to your concerns
>
> **Weakness 1.** For $\gamma$, we directly use $\gamma = 1$, which is the same as the original paper [1]. In this case, it is indeed not a hyperparameter for GPFL. For hyperparameter $\lambda$ and $\beta$, here is the sensitivity analysis results over dataset NCI1 with 25 clients and $\alpha = 0.5$
>
> |$\lambda$|performance|
> |---------|-----------|
> |1| 0.7640(0.013)     |
> |0.98| 0.7866(0.015)  |
> |**0.95**| 0.7888(0.012)|
> |0.92| 0.7866(0.014)|
> |0.9| 0.7912(0.015)   |
> |0.8| 0.7915(0.011)   |
>
> |$\beta$|performance|
> |---------|-----------|
> |1| 0.7852(0.011)     |
> |0.98| 0.7881(0.013)  |
> |**0.95**| 0.7888(0.012)|
> |0.92| 0.7888(0.015)|
> |0.9| 0.7890(0.015)   |
> |0.8| 0.7849(0.011)   |
>
> $\lambda = 1$ and $\beta = 1$ means that there are no self-connections and no initial graph updating, respectively. As the result shows, GPFL can achieve promising performance across different hyperparameter settings.
>
> __Requested Change 1.__ We will replace it with the following formulation: $$\theta^{t+1}\_i \xleftarrow{} \theta^t\_i + \mathop{AGGR}\_{j\in\mathbf{N}(i)}(\phi(\Delta\theta^t\_i,\Delta\theta^t\_j,w\_{ij})),$$ to clarify the confusion, where $\mathbf{N}(i)$ is the neighborhood of client $i$ and AGGR$ is the aggregation function that corresponds to the parameter aggregation process.
>
> **Requested Change 2.** We will add a pseudo-algorithm by Figure 2, which contains the formulation of marginal parameters.
>
> **Requested Change 3.** We compare the average time cost of FedAvg and GPFL on 1 communication round. The result is as follows:
> | Setting |Method| Time Cost|Method| Time Cost|
> |---------|------|----------|------|----------|
> |PROTEINS($\sharp client = 25$)|FedAvg| 0.404| GPFL |0.8341|
> |NCI1($\sharp client = 25$)|FedAvg | 0.706   | GPFL | 1.134 |
> |Yeast($\sharp client = 25$)|FedAvg| 1.989   | GPFL | 2.475 |
>
> The sizes of the datasets are 1113, 4110, and 19136. The table shows that, on average, GPFL is only 0.4 seconds slower than FedAvg. As the dataset grows larger, the proportion of this additional time cost further shrinks since local training consumes more time and GPFL only costs around 24% additional time on client graph learning and aggregation in the Yeast dataset. In real-world scenarios, the dataset can be even larger where GPFL would not cost too much time compared to other FL methods.
>
> You can also refer to Sec 4.5 for the runtime analysis of the whole FL process where we study the time cost of each component and compare the efficiency of GPFL and Self-train.
>
>
> [1] Zhitao Ying, Dylan Bourgeois, Jiaxuan You, Marinka Zitnik, and Jure Leskovec. Gnnexplainer: Generating explanations for graph neural networks. In NeurIPS, 2019.

---

### Author Response · Authors · 2025-06-25
**Sensitivity Analysis and Client Graph Analysis for all reviewers**

## 1. Sensitivity Analysis
We list sensitivity analysis results here as reviewers all mentioned this question.
As the number of clients varies across diffent settings, we set $$h = \frac{h'}{\sharp clients}$$
in our experiments, and we fixed $h' = 0.1$ in all experiments, which means we only cut off significantly weak connections between different clients. Following are the results on dataset NCI1 with 25 clients and $\alpha = 0.5,1,5$.

|$h'$ |Performance($\alpha = 0.5$)|Performance($\alpha = 1$)|Performance($\alpha = 5$)|
|----|-----------|------|-------|
|0.05 | 0.7873(0.012)| 0.7388(0.008)| 0.6889(0.02) |
|**0.1**|0.7888(0.012)| 0.7378(0.010)|0.6941(0.009)|
|0.2  | 0.7898(0.013) | 0.7325(0.009) |0.6890(0.017) |
|0.5| 0.7863(0.013) | 0.7364(0.005)| 0.6953(0.012) |
|1  | 0.7871(0.013) | 0.7388(0.009)| 0.6916(0.018)|
|2  | 0.7678(0.006) | 0.6892(0.005)| 0.6602(0.012)|

With a larger $h'$, GPFL is more likely to cut off weak links between clients. We choose a small $h' = 0.1$ to only filter significantly faint connections and encourage inter-client collabration. However, as the results show, GPFL is insensitive to the exact choice and maintains promising results as long as $h' \leq 1$.

## 2. Client Graph Analysis
To analyze the behavior of GPFL, we construct a "reference" client graph based on the label distribution of clients. we construct the reference graph by calculating the cosine similarity between the marginal label distribution (same way as our marginal parameters) of each client. Then, we utilize the Manhattan distance between the learned client graph $A$ and the label distribution graph $R$:$$L = Mean(|a_{ij} - r_{ij}|).$$
The following table shows how $L$ changes as the training process continues:
|Communication Round|$L$|
|-------------------|---|
|25| 0.0331|
|50| 0.0334|
|100| 0.0338|
|150| 0.0279|
|200| 0.0279|

As the result shows, the learned client graph $A$ maintains a close relationship to the reference graph $R$, which showcases that our framework is learning the latent distribution relationship between clients through the training process.

---

### Decision · Action_Editor_NERW · 2025-07-27

**Recommendation:** Accept as is

**Audience:**

Yes

**Audience Explanation:**

this is certainly true

**Claims And Evidence:**

Yes

**Claims Explanation:**

yes